# Flow and Heat Transfer Performances of Liquid Metal Based Microchannel Heat Sinks under High Temperature Conditions

**DOI:** 10.3390/mi13010095

**Published:** 2022-01-08

**Authors:** Tao Wu, Lizhi Wang, Yicun Tang, Chao Yin, Xiankai Li

**Affiliations:** Yangzhou Collaborative Innovation Research Institute Co., Ltd., Yangzhou 225006, China; lychee@mail.ustc.edu.cn (L.W.); tangyicun13@163.com (Y.T.); fortunatetomato@163.com (C.Y.); wootall62442@outlook.com (X.L.)

**Keywords:** microchannel heat sink, liquid metal, working fluid, cross-section shape, inlet velocity

## Abstract

Developments in applications such as rocket nozzles, miniature nuclear reactors and solar thermal generation pose high-density heat dissipation challenges. In these applications, a large amount heat must be removed in a limited space under high temperature. In order to handle this kind of cooling problem, this paper proposes liquid metal-based microchannel heat sinks. Using a numerical method, the flow and heat transfer performances of liquid metal-based heat sinks with different working fluid types, diverse microchannel cross-section shapes and various inlet velocities were studied. By solving the 3-D steady and conjugate heat transfer model, we found that among all the investigated cases, lithium and circle were the most appropriate choices for the working fluid and microchannel cross-section shape, respectively. Moreover, inlet velocity had a great influence on the flow and heat transfer performances. From 1 m/s to 9 m/s, the pressure drop increased as much as 65 times, and the heat transfer coefficient was enhanced by about 74.35%.

## 1. Introduction

In many industrial applications, such as rocket nozzles, miniature nuclear reactors and solar thermal generation, one of the major concerns is heat dissipation [1]. If the large amount heat cannot be removed quickly, the devices or systems cannot function or, worse, can be damaged. Hence, high-efficient heat transfer technology is important. In the aforementioned applications, the challenges of heat removal are attributed to high heat flux density, limited space for heat transfer equipment and accompanying high temperature. In this case, liquid metal-based microchannel heat sinks are well suited.

Microchannel heat sink has attracted worldwide attention from researchers because of its advantages, including high heat transfer coefficient, direct integration on the substrate and compactness. Since the pioneering work by Tuckerman and Pease [2], a lot of research has been conducted to study the thermal performance and hydraulic characteristics of various microchannel heat sinks. Wu and Little [3] tested silicon and glass microchannels with nitrogen, hydrogen and argon as the working fluids. The experimental data was fitted to give Nusselt number and friction factor correlations. Xu et al. [4] investigated the pressure drop characteristics in aluminum and silicon microchannels. It was found that the Poiseuille number remains constant in laminar flow region, as in conventional channels, but its value is different for different materials. The Poiseuille number was determined to be constant in laminar flow by Judy et al. [5] for fused silica and stainless steel microchannels using water, isopropanol and methanol as working fluids. Qu and Mudawar [6] manufactured a copper microchannel and tested it at different heat fluxes. The obtained pressure drop and temperature distribution were validated with a mathematical model developed by the authors, using the classical theory of conventional channels. Lee et al. [7] tested silicon microchannels with water and conclude that classical correlations can be applied to determine both the heat transfer and fluid flow characteristics of microchannel heat sinks. Liu and Garimella [8,9] conducted water flow tests on plexiglass microchannel samples, and found that the friction factor data are in agreement with conventional correlations for both laminar and turbulent flow. According to the research, it can be concluded that the single-phase fluid flow and heat transfer behaviors in microchannels are similar to those under normal scale and convectional theory is still appropriate to the flow characteristics [10,11].

In order to improve the cooling performance of microchannel heat sinks, the effect of geometrical factors including cross-sectional shape, microchannel pattern, manifolds and input/output ports have been widely studied [12]. The studies show that changing the geometrical factors could result in the optimization of average temperature, pressure drop, heat transfer coefficient, Nusselt number, as well as flow and temperature uniformity.

Jing and He [13] numerically studied the hydraulic and thermal performances of microchannel heat sinks with three different channel cross-sectional shapes: rectangle, ellipse and triangle. The calculating results show that the triangular-shaped microchannel has the smallest hydraulic resistance and the smallest convective heat transfer coefficient. Gunnasegaran et al. [14] studied the flow and convective heat transfer characteristics of water in rectangular, trapezoid and triangular shaped microchannels and found that for a given cross-sectional shape, the channel with smaller hydraulic diameter had larger convective heat transfer coefficient and pressure drop. Xia et al. [15] researched fluid flow in microchannels with different header shapes and inlet/outlet arrangements. They concluded that I-type had better flow uniformity than other cases due to the symmetrical flow distribution. Moreover, the effect of walls friction decreased the static pressure, which ultimately led to uniform velocity distribution. Kumar et al. [16,17] summarized the works focusing on the influence of inlet and outlet arrangements on the thermal and hydraulic performance. It was observed that vertical supply and collection system showed lower temperature non-uniformity and better liquid flow distribution.

Ahmed et al. [18] investigated the effect of geometrical parameters on laminar water flow and forced convection heat transfer characteristics in grooved microchannel heat sink. By means of numerical calculations, they found that with the optimum groove tip length ratio, groove depth ratio, groove pitch ratio and grooves orientation ratio, Nusselt number and friction factor could be improved by 51.59% and 2.35%, respectively. Zhu et al. [19] compared the thermal and hydraulic performance of wave microchannel heat sinks with wavy bottom ribs and side ribs design. The results showed that the up–down wavy design exhibited a better performance when small wavelengths were adopted. Wang et al. [20] proposed a new double layer wave microchannel heat sink and porous-ribs, they found that the porous-ribs exhibited obvious superiority at a low pumping power, and the wavy microchannel became domain at a high pumping power. Ermagan et al. [21] replaced the conventional walls with superhydrophobic walls in wave microchannel heat sinks to reduce the pressure loss. It was found that the beneficial effect of superhydrophobic walls on the comprehensive performance weakened as the waviness of the channel increased.

Gong et al. [22] designed a porous/solid compound fins microchannel heat sink to enhance the cooling performance. They found that the viscous shear stress decreased at the interfaces between fluid and porous fins, which resulted in the reduction of pressure drop. Ghahremannezhad et al. [23] proposed a new porous double-layered microchannel heat sink and found that the new microchannel heat sink showed not only good heat transfer performance, but also exhibited lower pressure drop. Li et al. [24] compared the thermal performance and flow characteristics of five heat sink designs, including porous-ribs single-layered, solid-ribs single-layered, solid-ribs double-layered, porous-ribs double-layered and mixed double-layered. It was found that the mixed double-layered microchannel heat sink processed a combination of low pressure drop and high thermal performance.

Besides geometric parameters, the proper selection of working fluid type is also a crucial factor to the performances of microchannel heat sinks [25,26]. Conventional cooling liquids, like water, could not meet the rapidly increasing demand of high-density heat dissipation because of its low thermal conductivity. In order to enhance the effective thermal conductivity, the method of adding conductive nano metal particles such as copper or aluminum to the solution and suspension is proposed by researchers. Jang and Choi [27] and Farsad et al. [28] showed numerically that water-based nanofluids enable microchannel heat sinks to dissipate heat fluxes as high as 1350–2000 W/cm^2^. Sohel et al. [29,30] showed analytically that 0.5–4.0 vol% CuO nanofluid flow in a copper microchannel heat sink having circular channels provides far better heat transfer performance and lower friction factor than Al_2_O_3_ and TiO_2_ nanofluids. Sivakumar et al. [31] also showed that CuO nanofluid provides better heat transfer coefficient enhancement than Al_2_O_3_ nanofluid. Salman et al. [32,33] showed numerically that dispersing SiO_2_ nanoparticles in ethylene glycol (base liquid) provides the highest Nu, followed by ZnO, CuO and Al_2_O_3_, and Nu increases with decreasing nanoparticles size. Kumar et al. [34] conducted thermofluidic analysis of Al_2_O_3_-water nanofluid cooled, branched wavy heat sink microchannels using a numerical method. The results showed that apart from disruption of the boundary layer and its reinitialization, vortices were formed near the secondary channel, which improved thermal performance. The heat transfer coefficient increased with increasing nanofluids concentrations for any investigated Reynolds number. Wang et al. [35] numerically investigated the forced convection in microchannel heat sinks using multi-wall carbon nanotube-Fe_3_O_4_ hybrid nanofluid as coolant working fluid. According to the results, the heat sinks, which consist of metallic foam, have better cooling performance and are able to decrease the surface temperature. However, the performance improvements brought by nanofluids are still limited due to the base fluids used. Particularly, the mixed solution may easily be subject to additional troubles such as susceptibility to fouling, particle deposition or conglomeration, degeneration of solution quality and flow jamming over the channels [36,37]. Besides, the low evaporation point of these liquids may imply potential dangers in preventing the device from burning out, since the liquids may easily escape to the ambient air.

Apart from nanofluids, liquid metal could also lead to an excellent cooling capacity because of its high thermal conductivity. In fact, liquid metals such as sodium, sodium-potassium alloy and lithium have been used as coolant in nuclear engineering for a long time. Nowadays, liquid metal cooling has been widely used in nuclear-powered plants with many different metals having been tried. Liquid metal cooling is also used in accelerators, solar power generation, LEDs and lasers [38,39,40]. Since Liu and Zhou [41] proposed for the first time to use low-melting-point liquid metals as an ideal coolant for the thermal management of computer chips, a lot of research has been performed to investigate the heat transfer capability. Miner and Goshal [36] carried out analytical and experimental work on liquid metal flow in a pipe. Their results indicated that the heat transfer is enhanced in both laminar and turbulent regimes using liquid metal coolant. Goshal et al. [42] used a GaIn alloy-based heat sink in a cooling loop and achieved a thermal resistance of 0.22 K/W for the entire system. Hodes et al. [43] studied the optimum geometry for water-based and Galinstan-based heat sinks in terms of minimum thermal resistance. It was shown that in the optimized configurations, Galinstan can reduce the overall thermal resistance by about 40% compared to water. Zhang et al. [44] carried out a follow-up work and according to their experimental data, liquid metal could enhance the convective heat transfer due to the superior thermophysical properties.

Currently, studies on liquid metal cooling or microchannel heat sink are widely conducted, but research on microchannel heat sinks using liquid metal as the working fluid is scarce. Some related investigations aim at dealing with the heat dissipation problem of computer chips or electronic devices at room temperature. However, in some applications, such as rocket nozzles, miniature nuclear reactors and solar thermal generation, high-density heat needs to be dissipated or transferred in a limited space under high temperature. In order to solve the problem of high-density heat dissipation under high temperature and limited space, this paper takes liquid metal-based microchannel heat sinks as object, establishing 3D physical and mathematical numerical models, so that flow and heat transfer performances with diverse working fluid types, various microchannel cross-section shapes and different inlet velocities could be obtained and analyzed.

## 2. Modeling and Numerical Methods

### 2.1. Physical Model

The structure of the investigated microchannel heat sink is shown in Figure 1. The cylindrical inlet and outlet passages are connected with 10 microchannels by inlet and outlet manifolds. In the cooling process, liquid metal enters the inlet passage and then flows into the microchannels through inlet manifold. After that, the liquid metal successively flows through the outlet manifold and outlet passages and finally flows out of the heat sink.

The overall size of the heat sink is *L* × *W* × *H* = 19 mm × 10 mm × 2 mm. The radius and height of inlet and outlet passages are 0.5 mm and 0.5 mm, respectively. Both the inlet and outlet manifolds are 3 mm long. All the microchannels have the same size of *L_c_* × *W_c_* × *H_c_* = 13 mm × 0.4 mm × 1 mm. Besides rectangular microchannel cross-section shape, three cross-section shapes (circle, trapezoid and parallelogram) are also studied in this paper. Different microchannel cross-section shapes are shown in Figure 2. The four different cross-section shapes have the same hydraulic diameter (*D_h_* = 0.57 mm).

In order to investigate the flow and heat transfer performances of the microchannel heat sink under high temperature conditions, silicon carbide (SiC) was applied for solid wall materials and alkalis with high melting point were selected as the liquid metal coolant. The thermo-physical properties of SiC were assumed to be constant. The density, specific heat capacity and thermal conductivity of SiC were 3220 kg/m^3^, 800 J/(kg·K) and 80 W/(m·K) respectively.

As for the alkalis, sodium (Na), potassium (K), sodium–potassium alloy (Na-K) and lithium (Li) were considered in this paper. For these alkalis, their thermo-physical property correlations as well as the range of validity of each correlation were given [45,46]. For all the correlations, temperature (*T*), density (*ρ*), specific heat capacity (*Cp*), thermal conductivity (*k*) and dynamic viscosity (*η*) were respectively given in K, kg/m^3^, J/(kg·K), W/(m·K) and Pa·s.

(1) Sodium (Na)

Density (370 K ≤ *T* ≤ 1100 K),
(1)$$\rho =T\left[T\left(0.9667\times 10^{-9}\cdot T-0.46005\times 10^{-5}\right)-0.1273534\right]+954.1579$$

Specific heat capacity (370 K ≤ *T* ≤ 1100 K),
(2)$$Cp=1630.14-0.4631\cdot \left(1.8T\right)+0.14284\times 10^{-3}\cdot {\left(1.8T\right)}^{2}$$

Thermal conductivity (370 K ≤ *T* ≤ 1100 K),
(3)$$k=93.9892-0.032503t_{f}+3.6197\times 10^{-6}\cdot t_{f}^{2}$$
(4)$$t_{f}=1.8\left(T-273.15\right)+32$$

Dynamic viscosity (370 K ≤ *T* ≤ 1100 K),
(5)$$\eta =\frac{0.11259\times 10^{-3}\cdot e^{\frac{749.08\rho }{1000T}}}{{\left(1000/\rho \right)}^{0.3333}}$$

(2) Potassium (K)

Density (334 K ≤ *T* ≤ 2270 K),
(6)$$\begin{array}{l} \rho =841.5-0.2172\cdot \left(T-273.15\right)-2.7\times 10^{-5}\cdot {\left(T-273.15\right)}^{2}\\ +4.77\times 10^{-9}\cdot {\left(T-273.15\right)}^{3} \end{array}$$

Specific heat capacity (373 K ≤ *T* ≤ 1423 K),
(7)$$Cp=4186.8\left[0.2004-0.8777\times 10^{-4}\cdot \left(T-273.15\right)+1.097\times 10^{-7}\cdot {\left(T-273.15\right)}^{2}\right]$$

Thermal conductivity (373 K ≤ *T* ≤ 1300 K),
(8)$$k=81.021-8.1812\times 10^{-2}\cdot T+4.2521\times 10^{-5}\cdot T^{2}-1.0589\times 10^{-8}\cdot T^{3}$$

Dynamic viscosity (334 K ≤ *T* ≤ 2000 K),
(9)ln(η)=−6.4846−0.42903·ln(T)+485.32/T

(3) Sodium-Potassium (NaK)

The density and specific heat capacity were defined on the basis of additive laws by means of weight fraction (*wt*) weighting. The dynamic viscosity was based on the mole fraction (*at*) weighting.

Density (273 K ≤ *T* ≤ 1573 K),
(10)$$\rho =\frac{1}{\frac{wt_{Na}}{\rho_{Na}}+\frac{wt_{K}}{\rho_{K}}}$$

Specific heat capacity (273 K ≤ *T* ≤ 1573 K),
(11)$$Cp=wt_{Na}\cdot Cp_{Na}+wt_{K}\cdot Cp_{K}$$

Thermal conductivity (273 K ≤ *T* ≤ 1573 K),
(12)$$k=15.0006+30.2877\times 10^{-3}\cdot T-2.08095\times 10^{-5}\cdot T^{2}$$

Dynamic viscosity (273 K ≤ *T* ≤ 1573 K),
(13)$$\eta =\frac{1}{\frac{at_{Na}}{\eta_{Na}}+\frac{at_{K}}{\eta_{K}}}$$

(4) Lithium (Li)

Density (455 K ≤ *T* ≤ 1500 K),
(14)$$\rho =278.5-0.04657T+274.6{\left(1-T/3500\right)}^{0.467}$$

Specific heat capacity (455 K ≤ *T* ≤ 1500 K),
(15)$$Cp=4754-0.925T+2.91\times 10^{-4}\cdot T^{2}$$

Thermal conductivity (455 K ≤ *T* ≤ 1500 K),
(16)$$k=22.28+0.05T-1.243\times 10^{-5}\cdot T^{2}$$

Dynamic viscosity (455 K ≤ *T* ≤ 1500 K),
(17)ln(η)=−4.16435−0.6374·ln(T)+292.1/T

### 2.2. Mathematical Model

#### 2.2.1. Governing Equations and Boundary Conditions

In order to establish the mathematical model of flow and heat transfer processes in the microchannel heat sink, the following assumptions are made in this paper:(1)Both the fluid flow and heat transfer are steady.(2)The fluid flow is incompressible and single phase.(3)There is no slip between fluid and wall.(4)Radiation heat transfer and viscous dissipation effect are neglected.

According to previous research [47], when the size of the microchannel is over 200 μm, the continuous medium hypothesis of fluid is still applicable. Hence, the governing equations for continuity, momentum and energy could be written as follows [48].

Continuity equation,
(18)$$\frac{\partial u}{\partial x}+\frac{\partial v}{\partial y}+\frac{\partial w}{\partial z}=0$$
where *u*, *v*, *w* in m/s are respectively the velocity in *x*, *y*, *z* direction

Momentum equations,
(19)$$u\frac{\partial u}{\partial x}+v\frac{\partial u}{\partial y}+w\frac{\partial u}{\partial z}=-\frac{1}{\rho }\frac{\partial p}{\partial x}+\frac{\eta }{\rho }\left(\frac{\partial^{2}u}{\partial x^{2}}+\frac{\partial^{2}u}{\partial y^{2}}+\frac{\partial^{2}u}{\partial z^{2}}\right)$$
(20)$$u\frac{\partial v}{\partial x}+v\frac{\partial v}{\partial y}+w\frac{\partial v}{\partial z}=-\frac{1}{\rho }\frac{\partial p}{\partial y}+\frac{\eta }{\rho }\left(\frac{\partial^{2}v}{\partial x^{2}}+\frac{\partial^{2}v}{\partial y^{2}}+\frac{\partial^{2}v}{\partial z^{2}}\right)$$
(21)$$u\frac{\partial w}{\partial x}+v\frac{\partial w}{\partial y}+w\frac{\partial w}{\partial z}=-\frac{1}{\rho }\frac{\partial p}{\partial z}+\frac{\eta }{\rho }\left(\frac{\partial^{2}w}{\partial x^{2}}+\frac{\partial^{2}w}{\partial y^{2}}+\frac{\partial^{2}w}{\partial z^{2}}\right)$$
where *p* in Pa is pressure, *ρ* in kg/m^3^ is density and *η* in Pa·s is viscosity.

Energy equations, in the fluid region,
(22)$$u\frac{\partial T}{\partial x}+v\frac{\partial T}{\partial y}+w\frac{\partial T}{\partial z}=\frac{\lambda_{f}}{\rho Cp}\left(\frac{\partial^{2}T}{\partial x^{2}}+\frac{\partial^{2}T}{\partial y^{2}}+\frac{\partial^{2}T}{\partial z^{2}}\right)$$
in the solid region,
(23)$$\frac{\partial^{2}T}{\partial x^{2}}+\frac{\partial^{2}T}{\partial y^{2}}+\frac{\partial^{2}T}{\partial z^{2}}=0$$
where *T* in K represents temperature, *λ_f_* in W/(m·K) is thermal conductivity and *Cp* in J/(kg·K) is specific heat capacity.

The calculating domain is divided into two parts: fluid domain and solid domain, as shown in Figure 3. The boundary conditions are set as follows. The velocity inlet and pressure outlet boundary conditions are applied for the inlet and outlet of the heat sink respectively. The constant heat flux boundary is set for the bottom surface. The coupled wall boundary conditions are applied for the interfaces between fluid domain and solid domain (microchannels’ surrounding surfaces, inlet and outlet manifold’s bottom surfaces). The thermal insulation wall boundary conditions are set for the other surfaces. To be clear, the ‘inlet velocity’ mentioned in this paper refers to the fluid velocity of the heat sink’s inlet, which is the set velocity of the velocity inlet boundary.

The commercial software ANSYS Fluent was used to perform the numerical calculations. When the calculations were complete, the velocity, pressure and temperature fields in solid and fluid domains of the heat sink could be obtained and the flow and heat transfer performances could be characterized by the following parameters.

Mean heat transfer coefficient *h_m_* [49],
(24)$$h_{m}=\frac{Q}{A_{w}\left(T_{w}-T_{f}\right)}$$
where *A_w_* is the heat transfer area between the fluid and the walls; *T_f_* and *T_w_* are the average temperature of the fluid and the walls respectively; *Q* is the heat exchange capacity which could be calculated as,
(25)$$Q=qA_{b}$$
where *q* and *A_b_* are the heat flux and area of the heat sink bottom surface.

Mean flow resistance coefficient *f* [50],
(26)$$f=\frac{\Delta P}{\left(\rho v^{2}/2\right)\left(L_{c}/D_{h}\right)}$$
where Δ*P* is the pressure drop, *ρ* and *v* are the density and velocity of the fluid.

#### 2.2.2. Mesh Independence and Model Validation

In order to ensure that the solutions are independent of grid size, three different grid systems with cell number of 1,627,410, 3,480,055 and 5,741,015 are generated and used to perfume the mesh independence study. Under the same boundary conditions, the fluid pressure drop calculated by the latter two grid systems varies only by 0.9%, as shown in Table 1. In order to save calculating time, the grid system with cell number of 3,480,055 is finally adopted in this paper.

Besides the mesh independence test, the validity of aforementioned numerical method was verified by means of theoretical results obtained by Mortensen et al. [51]. Figure 4 shows the comparison of hydraulic resistances between the theoretical and numerical results. The definition equation of the hydraulic resistance is,
(27)$$R_{H}=\frac{\Delta P}{V_{fr}}$$
where *R_H_* is the hydraulic resistance, Δ*P* is the pressure drop, *V_fr_* is the volume flow rate.

Compared to the theoretical values, the mean relative difference of the calculating values are 2.9%, 3.6% and 0.38% with liquid Na, liquid K and liquid Li, respectively; it can be concluded that the numerical results are agreeable with the existing theoretical results, which means the numerical method applied in this study has satisfactory accuracy.

The heat transfer calculating method is verified by comparing the result of Adeel Muhammad et al. [52]. Table 2 shows the comparison of total thermal resistance of the whole microchannel heat sink with the channel height varies from 3 mm to 9 mm. The maximum deviation is found to be less than 1%, which indicates the heat transfer calculating method applied in this paper is reliable.

The total thermal resistance of the heat sink is defined as,
(28)$$R_{Thm}=\frac{T_{\max}-T_{in}}{Q}$$
where *T*_max_ is the maximum temperature of the whole heat sink, *T_in_* is the inlet temperature of the liquid metal, *Q* is the heat exchange capacity.

## 3. Results and Discussions

### 3.1. Effects of Working Fluid

Seven alkalis are considered in this paper: sodium (Na), potassium (K), Na22-K78 alloy (0.22 weight fraction of Na, 0.78 weight fraction of K), Na44-K56, Na52-K48, Na56-K44 and lithium (Li).

Taking a rectangular microchannel cross-section, heat flux 200 W/cm^2^, inlet temperature 600 K and inlet velocity 3 m/s as typical condition, Figure 4 shows the temperature distributions of the heat sink with different working fluids (temperature contours have the same scale). It could be seen that taking Li as the working fluid has the best cooling effect while taking K as the working fluid has the worst. For the heat sinks with Na-K alloy as the working fluid, their cooling performances are intermediate between the Na-based and K-based heat sinks. Moreover, the bigger the weight fraction of Na, the better the cooling effect.

For all the heat sinks shown in Figure 5, the temperature gradient is mainly in the fluid flow direction. In the direction perpendicular to the fluid flow, the temperature distribution shows good uniformity, the working fluid and the heat sink base basically have the same temperature. Take the heat sink with Li as an example, Figure 6 shows the temperature and velocity distributions at the center section of microchannels (1.5 mm high from the bottom surface). It can be seen that the difference of temperature between fluid and solid regions is very small, which indicates the liquid alkali-based microchannel heat sink has excellent heat exchange ability.

Figure 7 shows the mean heat transfer coefficients and Nusselt numbers under various conditions with rectangular microchannel cross-section, inlet temperature 600 K and inlet velocity 3 m/s. For all the investigated alkalis, taking Li as the working fluid obtains the highest mean heat transfer coefficient while taking K as the working fluid gets the lowest. When the working fluid is Na, the mean heat transfer coefficient is the second highest, slightly lower than that when the working fluid is Li. Affected by the thermal conductivity, the changing tendency of Nusselt number is different with that of mean heat transfer coefficient. Liquid Na-K alloy-based heat sinks have bigger Nusselt numbers because of Na-K alloys’ smaller thermal conductivities. On the contrary, the relatively big thermal conductivity of Na leads to a small Nusselt number. Moreover, with the same working fluid, both Nusselt number and mean heat transfer coefficient vary slightly with the change of heat flux. This is because under different heat fluxes, the working fluid has different temperatures, which leads to the variation of thermo-physical properties as well as the heat transfer performances.

The convective heat transfer of liquid metals in microchannels depends on the liquid metals’ thermal properties. According to the convective heat transfer theory, Nusselt number, Nu, is associated with Reynold number, Re, and Prandtl number, Pr. The considered alkalis’ different thermo properties (thermal conductivity, specific heat capacity and viscosity) results in different Re and Pr, which leads various Nu as shown in Figure 7b. Moreover, based on the definition equation of *Nu*, the mean heat transfer coefficient can be calculated as,
(29)$$h_{m}=\frac{Nu\cdot k}{D_{h}}$$

Influenced by thermal conductivity, the relative magnitude of mean heat transfer coefficient among the investigated alkalis is different with that of *Nu*. Because of bigger thermal conductivities, the mean heat transfer coefficients of liquid Li and Na are obviously higher than liquid K and liquid Na-K alloys.

Figure 8 shows the pressure drops and flow resistance coefficients under various conditions with rectangular microchannel cross-section, inlet temperature 600 K and inlet velocity 3 m/s. It could be seen that the pressure drop and the flow resistance coefficient changes in different degrees with the change of heat flux, among which the biggest change happens when the working fluid is K. This is because the changes of K’s temperature and thermo-physical properties are the greatest. Furthermore, the flow resistance coefficient among all the investigated alkalis has no great difference, while the pressure drop is significantly smaller when the working fluid is Li. This is because the pressure drop is positively associated with the flow resistance as well as the working liquid’s density, as shown in Equation (26). Li’s density is the smallest among the considered liquid metals, which leads to the smallest pressure drop. Considering flow and heat transfer performances synthetically, Li is the optimum working fluid among all the investigated alkalis.

For comparative analysis, pumping power (pressure drop) should be kept consistent [53,54,55,56]. Therefore, in order to obtain the optimum working fluid, the mean heat transfer coefficient, *h_m_*, is compared under the same pressure drop. Figure 9 shows the changing trend of *h_m_* with different pressure drop at rectangular microchannel cross-section, inlet temperature 600 K, heat flux 200 W/cm^2^. Obviously, since using liquid Li could get the biggest heat transfer coefficient, it is the optimum working fluid among all the investigated alkalis.

### 3.2. Effects of Microchannel Cross-Section

In order to study the effects of microchannel cross-section on the flow and heat transfer performances, four microchannel cross-section shapes (rectangle, circle, trapezoid and parallelogram, as shown in Figure 2) are considered in this paper. Figure 10 shows temperature distributions of the heat sink with different microchannel cross-section shapes when working fluid is Li, heat flux is 200 W/cm^2^, inlet temperature is 600 K and inlet velocity is 3 m/s. Obviously, the temperature distribution is hardly influenced by the change of microchannel cross-section shape.

Figure 11 shows the mean heat transfer coefficients and Nusselt numbers under various conditions with working fluid Li, inlet temperature 600 K and inlet velocity 3 m/s. For the Nusselt number, its value depends on the channel geometry [13,50]. Within the study scope shown in Figure 11, using rectangular, trapezoidal or parallelogram cross-section shaped microchannel has basically the same Nusselt number, while using circular microchannel could obtain significantly higher Nusselt number. This is because under the same hydraulic diameter, the area of circular cross-section is smaller than that of rectangular, trapezoidal and parallelogram cross-section, hence, the working fluid has bigger Reynold number inside the circular microchannel, which leads to the higher Nusselt number. According to the equation, *h_m_ =* Nu·*k*/*D_h_*, the variation of the mean heat transfer coefficient is accordance with Nusselt number. So, as shown in Figure 11a, using a circular cross-section obtains the highest heat transfer coefficient. Compared to the other three cross-section shapes, using circular cross-section could increase the mean heat transfer coefficient by about 14,000 W/(m^2^·K), which indicates the heat sink with circle microchannel cross-section has the best flow performance.

Figure 12 shows the pressure drops and flow resistance coefficients under various conditions with working fluid Li, inlet temperature 600 K and inlet velocity 3 m/s. Obviously, the heat sink with parallelogram cross-section has the lowest pressure drop and flow resistance coefficient while the heat sink with circular cross-section has the highest. This is because with the same hydraulic diameter, the investigated four microchannel cross-section shapes have different cross-sectional area, which leads to different velocity magnitudes as well as the pressure drop. Among the four different kinds of microchannels, the circular microchannel’s cross-sectional area is the smallest. Hence, under the same heat sink’s inlet velocity, the working fluid within the circular microchannel has the highest speed (as shown in Figure 13), which leads to the highest pressure drop.

In order to find the best cross-section shape, the performance evaluation criteria *PEC,* which represents the comprehensive performance of flow and heat transfer, is calculated. The *PEC* is defined as [57],
(30)$$PEC=\frac{Nu/Nu_{0}}{{\left(f/f_{0}\right)}^{1/3}}$$
where *Nu*_0_ and *f*_0_ are the benchmark Nusselt number and mean flow resistance coefficient. In this paper, *Nu* and *f* of the heat sink with rectangular microchannel cross-section under heat flux 100 W/cm^2^ are appointed as *Nu*_0_ and *f*_0_.

Figure 14 shows the *PEC* values of heat sinks with different microchannel cross-section shapes. Obviously, using circular microchannel could obtained the highest *PEC* values, indicating circle is the best choice for microchannel cross-section shape.

### 3.3. Effects of Inlet Velocity

Five inlet velocities (1 m/s, 3 m/s, 5 m/s, 7 m/s and 9 m/s) are considered in this paper so that the effects of inlet velocity on the flow and heat transfer performances could be investigated. Figure 15 shows temperature distributions of the heat sink with different inlet velocities under rectangular microchannel cross-section, heat flux 200 W/cm^2^, inlet temperature 600 K and Li as the working fluid. It could be seen that the inlet velocity has a significant effect on the temperature distribution. With the increase of inlet velocity, the temperature of the heat sink becomes obviously lower. Under the inlet velocity of 1 m/s, the high temperature region covers most of the heat sink, indicating the heat sink’s poor cooling effect. When the inlet velocity increases to 3 and 5 m/s, the high temperature region becomes much smaller. When the inlet velocity reaches 7 and 9 m/s, the entire heat sink is basically at a relatively low temperature level.

Figure 16 shows the mean heat transfer coefficients and Nusselt numbers under various conditions with working fluid Li, rectangular microchannel cross-section and inlet temperature 600 K (using pressure drop under corresponding inlet velocity as x-axis label so that the heat transfer performance could be compared properly [53,54,55,56]). For all the investigated heat fluxes, both the heat transfer coefficient and Nusselt number increase with the increase of inlet velocity. From 1 m/s to 9 m/s, the heat transfer coefficient and Nusselt number, respectively, enhance 74.35% and 80.5% at most. Moreover, the increasing rates of the heat transfer coefficient and Nusselt number are considerably high at low inlet velocity, but quite low at high inlet velocity. This implies that under small inlet velocity, the cooling ability of the heat sink could be improved by increasing inlet velocity. However, when the inlet velocity is high enough, continuing to increase the inlet velocity brings little benefit.

Figure 17 shows the pressure drops and flow resistance coefficients under various conditions with working fluid Li, rectangular microchannel cross-section and inlet temperature 600 K. It could be seen that the pressure drop rises rapidly with the increase of inlet velocity. From 1 m/s to 9 m/s, the pressure drop increases as much as 65 times. As for the flow resistance coefficient, it decreases gradually with the increase of the inlet velocity. This is because although the flow resistance coefficient is proportional to the pressure drop (Δ*P*), it is inversely related to the square of velocity (*v*^2^) (according to Equation (26)).

## 4. Conclusions

Numerical investigations of flow and heat transfer performances in liquid metal-based microchannel heat sinks are presented in this paper. During the simulation processes, different working fluid types, microchannel’s cross-sectional geometries and inlet velocities are considered so that their influences on the heat sink’s characteristics can be analyzed. Based on the calculating results, some conclusions were drawn as follow:(1)Among all the seven investigated alkalis, lithium is the best option for working fluid because the lithium-based microchannel heat sink has the best cooling ability and the lowest pressure drop.(2)For the four considered microchannel cross-section types (rectangle, circle, trapezoid and parallelogram), utilizing a circular microchannel cross-section obtains a higher mean heat transfer coefficient, while using a parallelogram obtains the lowest pressure drop. Considering flow and heat transfer performances comprehensively, the circle is the optimum choice for microchannel cross-section shape because using a circular microchannel has the highest PEC value.(3)Inlet velocity has a significant influence on the heat sink’s flow and heat transfer performances. When the inlet velocity rises from 1 m/s to 9 m/s, the heat transfer coefficient enhances 74.35% at most, while the pressure drop increases up to 65 times. In order to obtain a favorable overall performance, the inlet velocity should be selected carefully.

## Figures and Tables

**Figure 1 micromachines-13-00095-f001:**
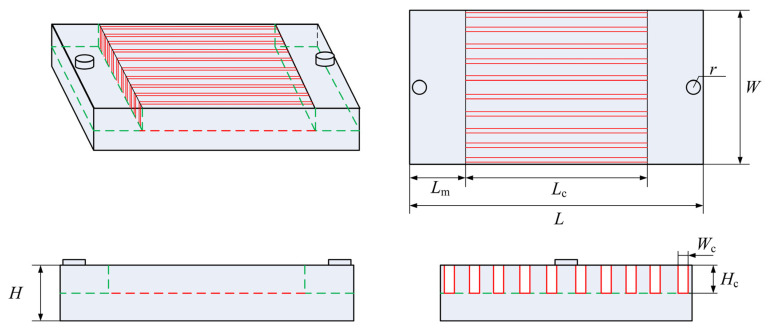
The structure of microchannel heat sink.

**Figure 2 micromachines-13-00095-f002:**
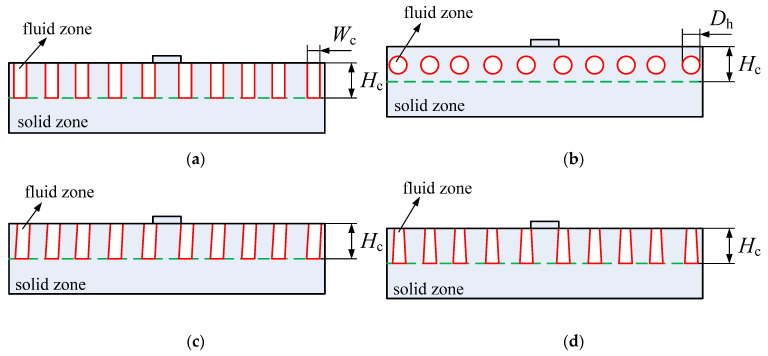
Diagram of different microchannel cross-section types. (**a**) Rectangle; (**b**) circle; (**c**) parallelogram; (**d**) trapezoid.

**Figure 3 micromachines-13-00095-f003:**
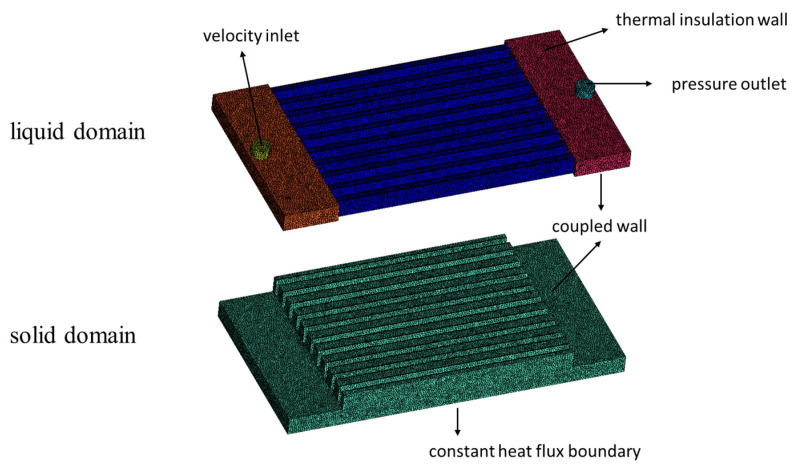
Computational domain and boundary conditions.

**Figure 4 micromachines-13-00095-f004:**
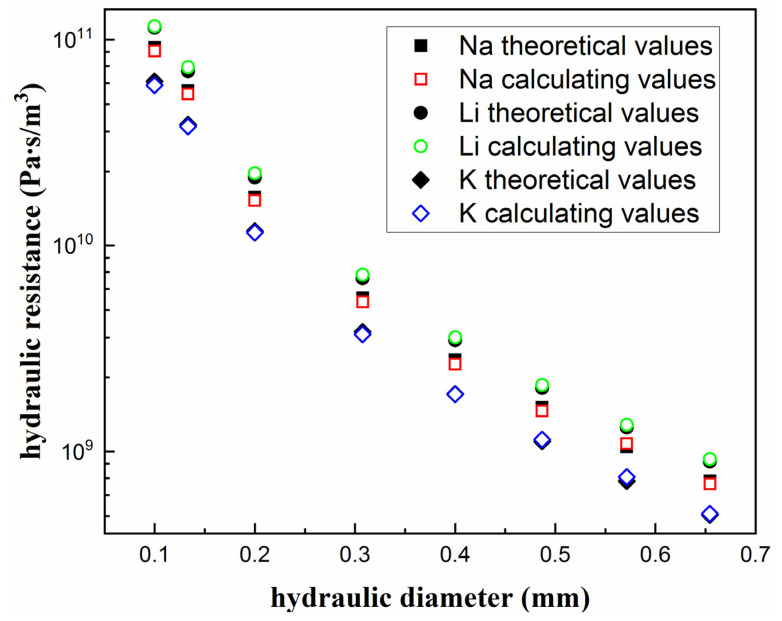
Comparison between theoretical and numerical results of dimensionless hydraulic resistance.

**Figure 5 micromachines-13-00095-f005:**
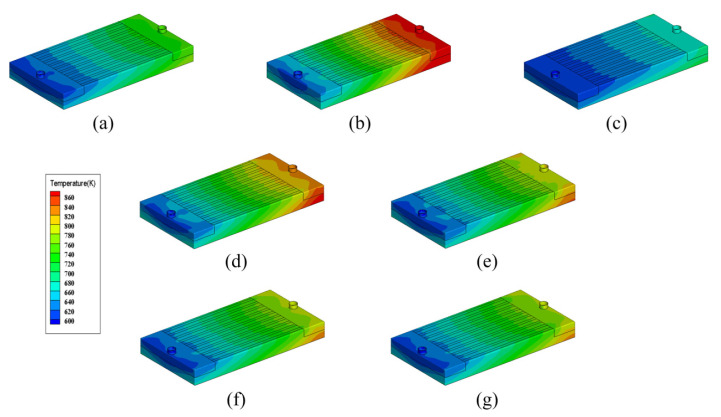
Temperature distributions of the heat sink with different working fluids. (**a**) Na; (**b**) K; (**c**) Li; (**d**) Na22-K78; (**e**) Na44-K56; (**f**) Na52-K48; (**g**) Na56-K44.

**Figure 6 micromachines-13-00095-f006:**
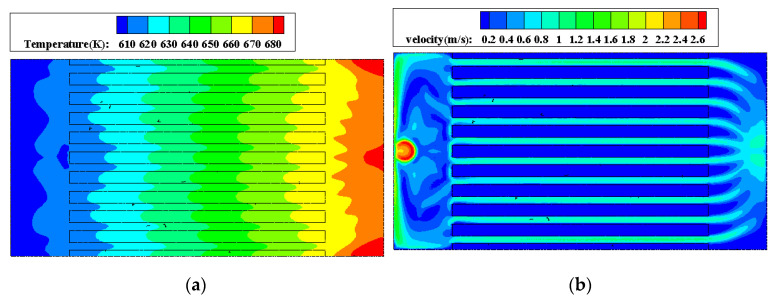
Temperature and velocity distributions at the center section of microchannels. (**a**) Temperature distribution; (**b**) velocity distribution.

**Figure 7 micromachines-13-00095-f007:**
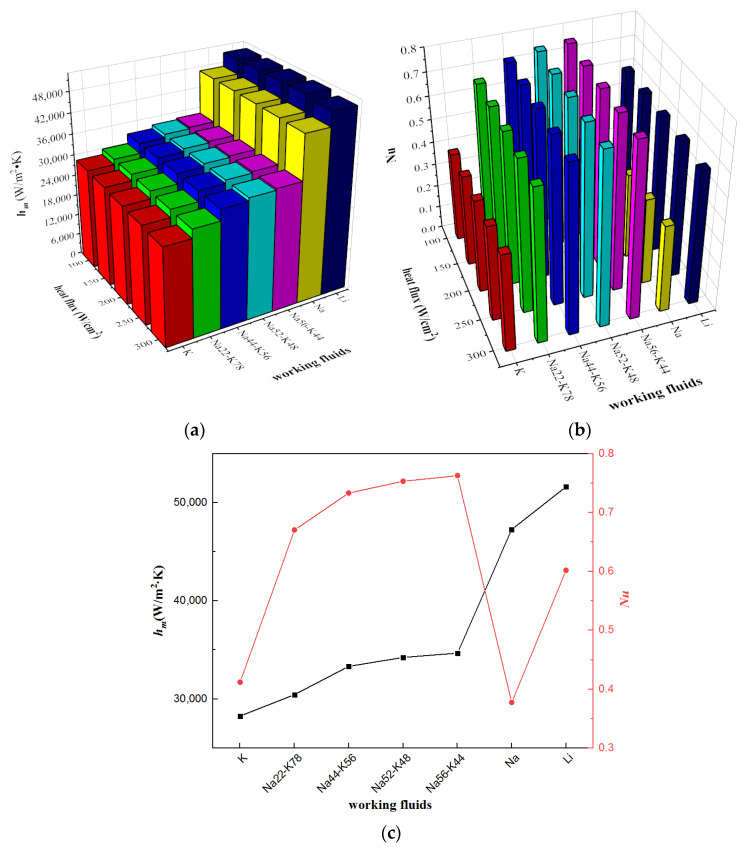
Mean heat transfer coefficients and Nusselt numbers with different working fluids. (**a**) Mean heat transfer coefficient; (**b**) Nusselt number; (**c**) Mean heat transfer coefficient and Nusselt number under heat flux 200 W/cm^2^.

**Figure 8 micromachines-13-00095-f008:**
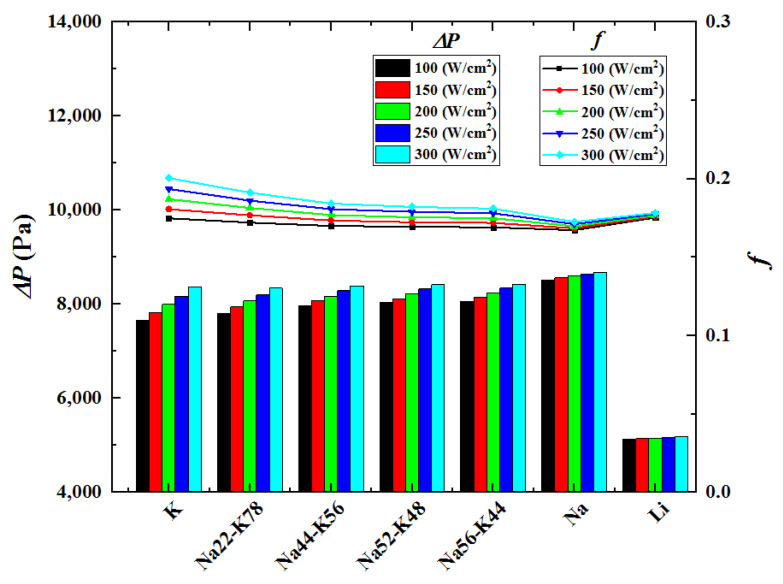
Pressure drops and flow resistance coefficients with different working fluids.

**Figure 9 micromachines-13-00095-f009:**
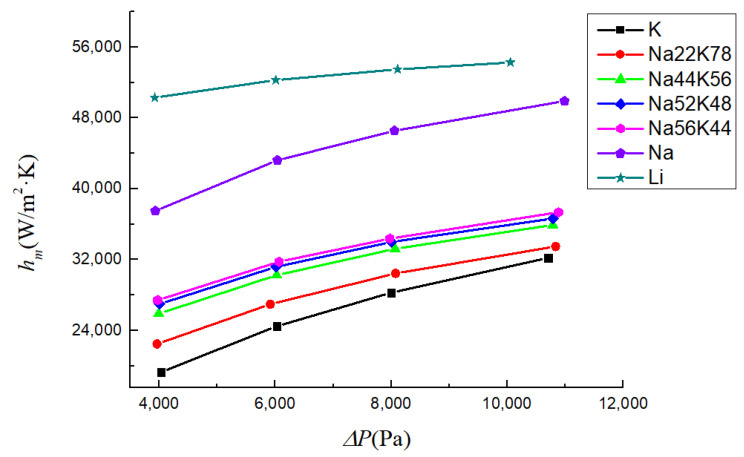
Variation of mean heat transfer coefficient (*h_m_*) as a function of pressure drop (Δ*P*).

**Figure 10 micromachines-13-00095-f010:**
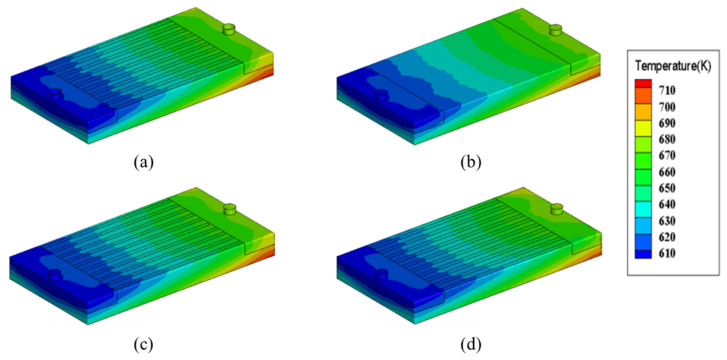
Temperature distributions of the heat sink with different microchannel cross-section types. (**a**) Rectangle; (**b**) circle; (**c**) trapezoid; (**d**) parallelogram.

**Figure 11 micromachines-13-00095-f011:**
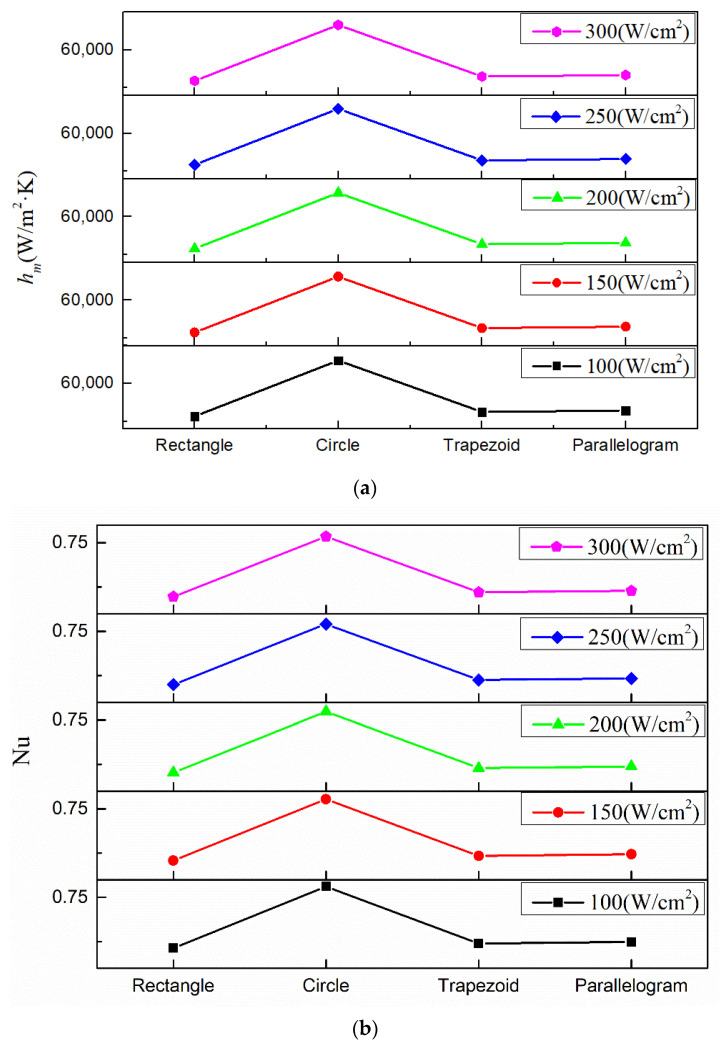
Mean heat transfer coefficients and Nusselt numbers with different micro channel cross-section types. (**a**) Mean heat transfer coefficient; (**b**) Nusselt number.

**Figure 12 micromachines-13-00095-f012:**
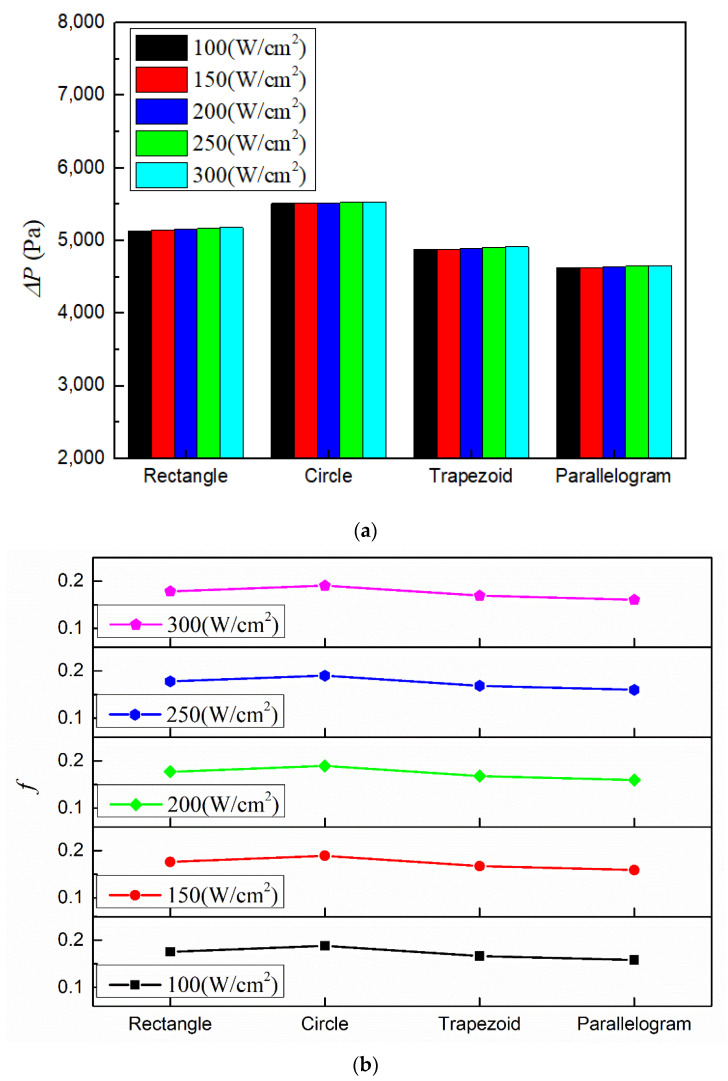
Pressure drops and flow resistance coefficients with different microchannel cross-section types. (**a**) Pressure drop; (**b**) flow resistance coefficient.

**Figure 13 micromachines-13-00095-f013:**
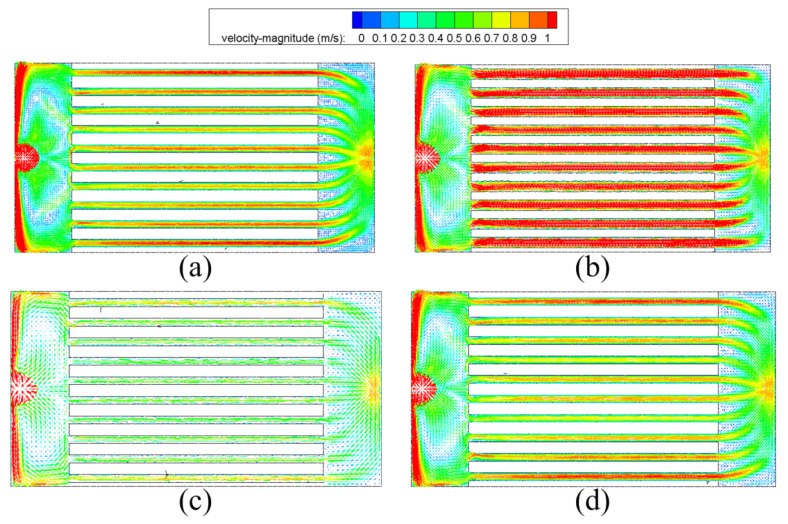
Velocity vectors at the center section of the heat sink with different microchannel cross-section types. (**a**) Rectangle; (**b**) circle; (**c**) trapezoid; (**d**) parallelogram.

**Figure 14 micromachines-13-00095-f014:**
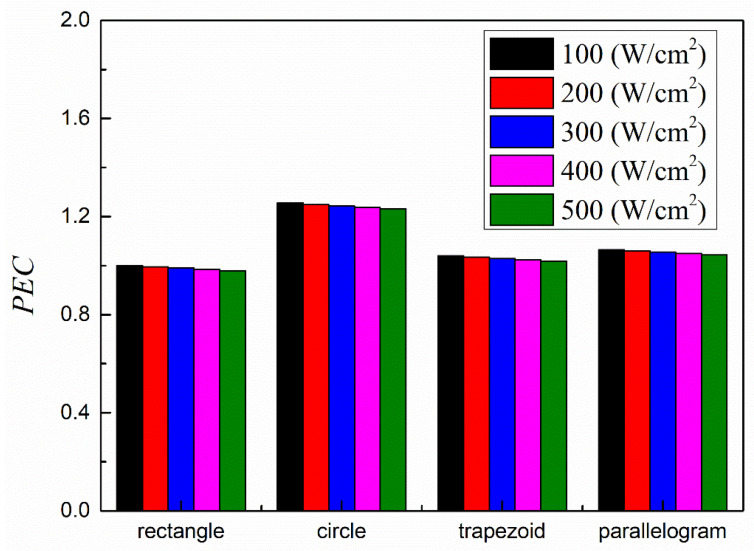
*PEC* of heat sinks with different microchannel cross-section shapes.

**Figure 15 micromachines-13-00095-f015:**
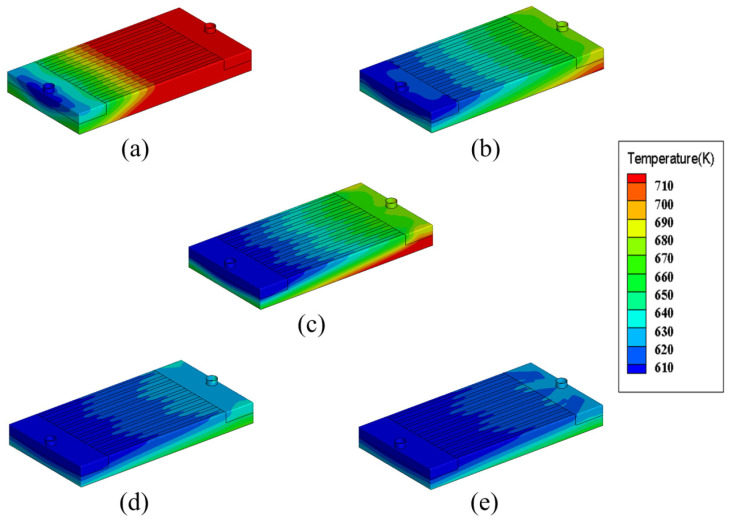
Temperature distributions of the heat sink with different inlet velocities. (**a**) 1 m/s; (**b**) 3 m/s; (**c**) 5 m/s; (**d**) 7 m/s; (**e**) 9 m/s.

**Figure 16 micromachines-13-00095-f016:**
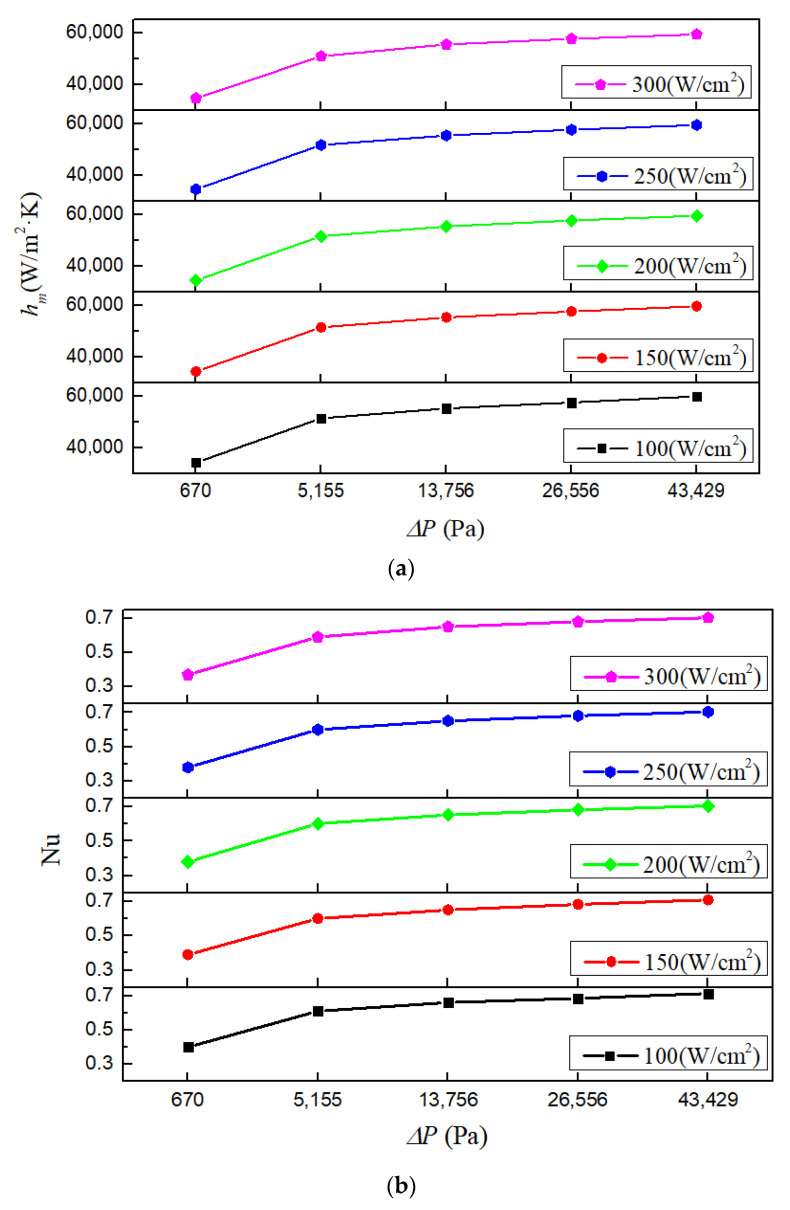
Mean heat transfer coefficients and Nusselt numbers with different inlet velocities. (**a**) Mean heat transfer coefficient; (**b**) Nusselt number.

**Figure 17 micromachines-13-00095-f017:**
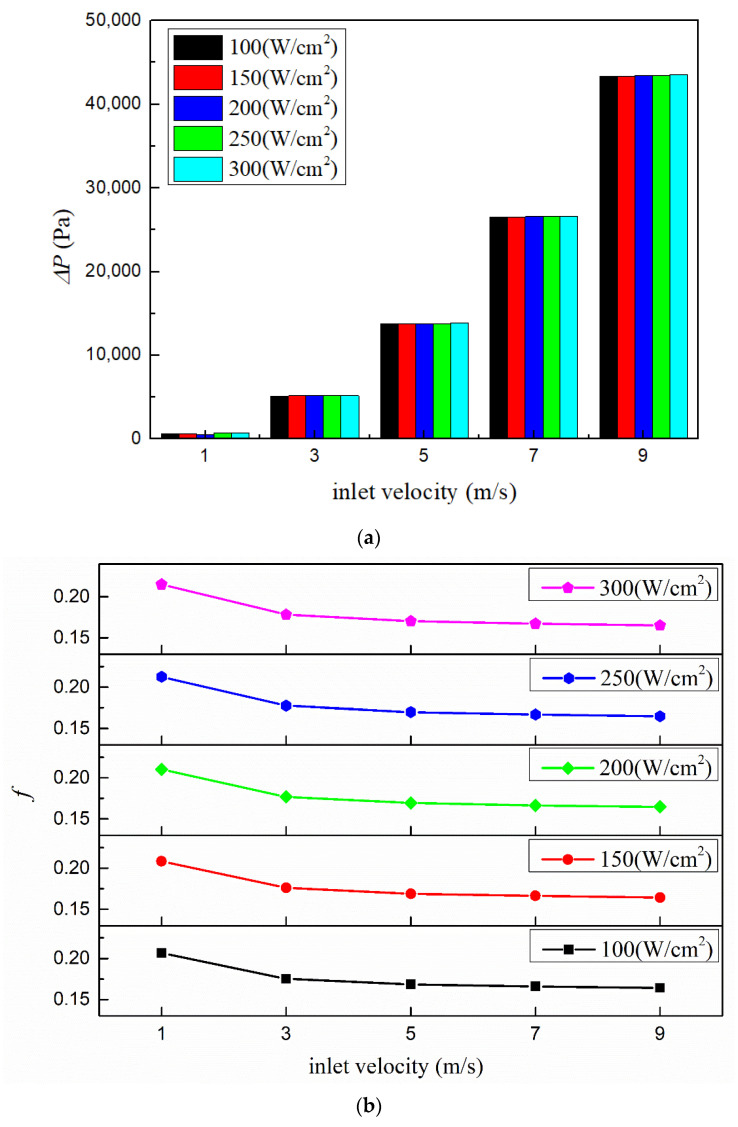
Pressure drops and flow resistance coefficients with different inlet velocities. (**a**) Pressure drop; (**b**) flow resistance coefficient.

**Table 1 micromachines-13-00095-t001:** Mesh independency analysis.

Cell Number	Calculated Pressure Drop	Relative Difference
1,627,410	9709.92 Pa	/
3,480,055	10,320.50 Pa	5.9%
5,741,015	10,414.30 Pa	0.9%

**Table 2 micromachines-13-00095-t002:** Comparison of total thermal resistance.

Microchannel Height	*R_Thm_* (K/W) Present	*R_Thm_* (K/W) Ref. [52]	Relative Difference
3	0.1174	0.1184	0.84%
4	0.09944	0.1002	0.76%
5	0.09076	0.09156	0.87%
6	0.08648	0.08718	0.80%
7	0.08429	0.08478	0.58%
9	0.08282	0.08346	0.77%

## Data Availability

The data presented in this study are available on request from the corresponding author. The data are not publicly available due to privacy.

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
