# Peer review of "Flow and Heat Transfer Performances of Liquid Metal Based Microchannel Heat Sinks under High Temperature Conditions"

_micromachines, 2022, doi:10.3390/mi13010095_

Round 1
Reviewer 1 Report
Journal: Micromachines
Title of Article: Flow and Heat Transfer performances of liquid metal based micro-channel heat sinks under high temperature conditions
Manuscript ID: micromachines-1471000
Major comments:
The current investigations deal with the flow and
heat transfer performance of heat sink with different working fluids and micro-channel
cross section shapes. Moreover, the influences of inlet velocity on the cooling
ability and flow characteristic are analyzed. In my opinion, this work is interesting and appropriate for publication in the “Micromachines“ but some important points are recommended to be revised before publication.
Recommendation: Minor Revision
Comments:
The paper contains new and significant information adequate to justify publication. Moreover, it is well prepared, but a particular interest must be paid to these points:
- The introduction should clearly state what it is expected to do with the present studied. As it is too vague and a non-focus application of data is delivered, which must definitively be improved.
- In the literature survey part, a small number of previous studies have been mentioned and the discussion does not reveal the coherence among them. It now looks like a simple list of the abstracts. Also, the review must contain very recent papers deals with considered problem and must be more extended.
- Avoid lumped references in the introduction part, try to cite each reference separately for more clarity to the reader.
- This paper is well prepared, but physical meaning must be added. Why was this physical configuration selected? Is there any special meaning in applications?
- Boundary conditions are not well explained in the paper.
- Where is the mesh independency analysis figure or table?? Authors must prove the mesh independency through figures or tables for more credibility to the reader.
- In the validation part, authors should explain more the parameter used to compare results and extend more the difference discussion.
- In mathematical model section, add relevant references for the equations.
- The discussion of heat transfer rates through different working fluids in Fig.6 should be more discussed and physical phenomena explained.
- Many parameters used in the paper are missing in the nomenclature part.
- The English used in the whole paper need a proofreading and correction for many typos and grammatical errors.
Reviewer 2 Report
(1) The definition and the calculation method of the dimensionless hydraulic resistances shown in Fig. 3 should be given.
(2) It was found that, taking Li as the working fluid could obtain the highest mean heat transfer coefficient. The reason should be discussed.
(3) It was found that, the pressure drop is significantly smaller when the working fluid is Li. The reason should be discussed.
(4) It was found that, using trapezoidal and parallelogram cross section could always obtain the highest values for the mean heat transfer coefficient. The reason should be discussed.
(5) To verify the numerical method presented in this paper, a comparison with the theoretical or experimental results when the liquid metal is used as working fluid should be carried out.
Reviewer 3 Report
The manuscript models different liquid metal based working fluids as single phase in a heat sink by varying the channel cross-sectional shapes. However, the the manuscript has following shortcomings:
The manuscript is poorly written, the results are badly presented, and does not report any significant research findings.
Abstract needs to be improved a lot. This is obvious, " Moreover, inlet velocity has a great influence on the flow and heat transfer performances. With the raise of inlet velocity, both the pressure drop and heat transfer coefficient increase significantly."
Figure 2(b): What view of the heat sink with circular channel shape is presented here?
Why the selected liquid metal working fluids show variable thermal hydraulic performance, discuss the underlying physical phenomenon instead of just reporting the data in graphs.
The abstract and conclusion should have different content and writing style. Avoid repetition.
Round 2
Reviewer 1 Report
The authors have well addressed the comments so the manuscript can be accepted in the current form.
Author Response
Thanks for the reviewer's rigorous and careful comments.
Reviewer 2 Report
The following concerns are not answered in the revised version.
(1) It was found that, using trapezoidal and parallelogram cross section could always obtain the highest values for the mean heat transfer coefficient. The reason should be discussed.
(2) Only the flow resistant was verified in Section 2.2.2. The heat transfer performance should also be verified.
Author Response
Please see the attachment。

Reviewer 3 Report
The manuscript needs following improvements before it can be considered for publication.
- The manuscript needs major improvements in presentation of results.
- Highlight/mention the fluid zone in Figure 2.
- No variation of heat transfer coefficient and Nusselt number is visible as the data is presented in Figure 7.
- What are these figures (7, 10b, 11b, 13, 14b) supposed to illustrate? Use a single scale to compare the data of different cases.
- Figure 5 & 12, one color map is sufficient.
- Fig. 11a, why the pressure drop is higher with circular channel cross-section? Show flow streamlines/vectors to demonstrate the flow behavior around different channel cross-sections.
Round 3
Reviewer 2 Report
It was stated that the working fluid has bigger Reynold number inside the circular micro-channel. However, since the inlet velocity and the hydraulic diameter are the same for different micro-channel cross section types, the Reynold number should be the same.
Reviewer 3 Report
“Using circular microchannel could obtain significantly higher Nusselt number. This is because under the same hydraulic diameter, the area of circular cross section is smaller than that of rectangular, trapezoidal and parallelogram cross section, hence, the working fluid has bigger Reynold number inside the circular micro-channel, which leads to the higher Nusselt number.”
For comparative analysis, the pumping power (pressure drop) should be kept consistent. Since, density and thermal viscosity of each working fluid is different, plotting the results against inlet velocity does not lead to true comparison of heat transfer coefficient and Nusselt number of different working fluids. Therefore, a dimensionless parameter (Reynolds number) should be used instead of inlet velocity for comparison. Review the papers below for reference.
- Performance analysis of hybrid nanofluid in a heat sink equipped with sharp and streamlined micro pin-fins
- Pin-fin shape-dependent heat transfer and fluid flow characteristics of water- and nanofluid-cooled micropin-fin heat sinks: Square, circular and triangular fin cross-sections
- Numerical analysis of the heat transfer and fluid flow characteristics of a nanofluid-cooled micropin-fin heat sink using the Eulerian-Lagrangian approach
- Analysis of hydro-thermal and entropy generation characteristics of nanofluid in an aluminium foam heat sink by employing Darcy-Forchheimer-Brinkman model coupled with multiphase Eulerian model
Author Response
Thanks for the reviewer’s comment. The comparison of heat transfer coefficient and Nusselt number at the same pressure drop has been added in the manuscript, and the mentioned studies have been referred.
Round 4
Reviewer 3 Report
It is recommended that the author spends proper time to understand and address each and every comment.
1. As mentioned earlier, change the x-axis label from inlet velocity to pressure drop in all graphs (Fig. 15a and 15b) for a sound comparative analysis.
2. 2nd conclusion: "Considering flow and heat transfer performances comprehensively, circle is the optimum choice for micro-channel cross section shape because using circular microchannel has the highest PEC value."
The author should add graph showing the PEC
Author Response
Corresponding revisions have been made in the manuscript according to the reviewer's comments.